# Spatial–Temporal Characteristics of Carbon Emissions in Mixed-Use Villages: A Sustainable Development Study of the Yangtze River Delta, China

Yiqun Wu [1,2,3], Yuan Sun [4,5], Congyue Zhou [3], Yonghua Li [3,6,7], Xuanli Wang [1] and Huifang Yu [1,2,*]

1   College of Art and Archaeology, Hangzhou City University, Hangzhou 310058, China; wuyq@hzcu.edu.cn (Y.W.); xliwang9020@163.com (X.W.)
2   Beautiful Hangzhou Environmental Planning and Architectural Design Research Center, Hangzhou City University, Hangzhou 310058, China
3   College of Civil Engineering and Architecture, Zhejiang University, Hangzhou 310027, China; 12012015@zju.edu.cn (C.Z.); lyh@zju.edu.cn (Y.L.)
4   Department of Earth and Environmental Sciences, The University of Manchester, Manchester M13 9PY, UK; 22012137@zju.edu.cn
5   China Institute of Urbanization, Zhejiang University, Hangzhou 310058, China
6   Center for Balance Architecture, Zhejiang University, Hangzhou 310058, China
7   Architectural Design and Research Institute of Zhejiang University Co., Ltd., Hangzhou 310028, China
*   Correspondence: yuhf@hzcu.edu.cn

**Abstract:** With the progression of novel urbanization, rural regions are increasingly characterized by mixed-use features, where work and living activities intersect, resulting in a significant surge in per capita carbon emissions. This research article aims to elucidate the spatio-temporal relationship of carbon emissions in rural areas and their association with mixed-use intensity from a sustainable development perspective. For the study, we selected four of the most representative mixed-use village types in the Yangtze River Delta region. Using the STING method, each rural space was delineated into micro-level mixed-use units. Subsequently, a quantitative evaluation model was constructed to gauge the relationship between mixed-use intensity and carbon emissions. This was complemented by employing GIS simulations to analyze the spatio-temporal attributes of carbon emissions in mixed-use villages. Our findings indicate that (1) different types of villages display considerable disparities in mixed-use intensity and carbon emissions. Their correlation also varies significantly, with traditional agricultural villages exhibiting the lowest values of 0.338 and 0.356, while E-commerce-centric villages recorded the highest at 0.674 and 0.653. (2) The carbon emissions of rural units manifest diverse patterns that include dispersed distribution, core aggregation, linear decay, and dissipative fragmentation. These correspond to traditional agriculture, industrial production, tourism service, and E-commerce villages, respectively. (3) The carbon emissions of mixed-use villages exhibit cyclical fluctuations over time, with different magnitudes observed across villages. Traditional agricultural villages display the smallest fluctuations (within 30%), while those centered around tourism services can experience fluctuations exceeding 150%. Building on these insights, we delved deep into the challenges faced by each village type in enhancing the quality of work and living while concurrently achieving energy conservation and emission reduction. Based on these aspects, we propose a sustainable low-carbon development pathway tailored for mixed-use villages.

**Keywords:** mixed-use villages; carbon emissions; spatial–temporal characteristics; sustainable development; Yangtze River Delta region

## 1. Introduction

The United Nations Environment Programme (UNEP) Global Emissions Report warns that to achieve the global temperature control target of 1.5 °C between 2020 and 2030, carbon emissions should be reduced by more than 7.6% per year [1]. Statistical reports

have revealed that China's carbon emissions have significantly increased from 1.50 tons per capita in 1980 at the beginning of the reform and opening-up to 9.52 tons per capita in 2022, with an average annual growth rate of 5.08% [2]. The rapid economic growth of nearly 10% annually has come at the expense of the environment, making China surpass the United States as the world's largest carbon-emitting country [3].

In 2020, China set a goal to reach peak carbon emissions by 2030 and achieve carbon neutrality by 2060, known as the "Dual Carbon" targets [4]. The establishment of a series of targets has expedited China's carbon emission control efforts and the implementation of reduction plans, necessitating coordinated collaboration across the nation. As a predominantly agricultural country, rural energy issues relate to the production and domestic energy consumption of nearly half of China's population [5]. Rural energy provides essential material foundations for production and living, and its consumption contributes to the annual growth in China's carbon emissions. According to the China Energy Statistical Yearbook [6], rural energy consumption has increased from 201 million tons of standard coal in 2014 to 311 million tons in 2020, accounting for approximately 48.3% of the country's domestic energy use. Carbon emissions rose from 889 million tons in 1979 to 3.43 billion tons in 2020, representing 43.52% of China's total emissions. During this period, the per capita carbon emission growth in rural areas was 2.4 times that of urban locales [7], positioning the rural regions as a pivotal segment in China's pursuit of the "Dual Carbon" objectives. China has introduced a series of policies related to eco-agricultural development [8], green rural housing construction [9], clean energy promotion [10], and rural domestic wastewater treatment [11], indicating that the control of rural carbon emissions has been placed on the agenda.

In the context of new urbanization, the land development and utilization of central towns have undergone a transition from early aggressive growth to the exploration of existing resources and the limitation of incremental expansion [12]. Consequently, rural areas have emerged as spatial carriers that accommodate the spillover functions of cities. A cluster of well-developed villages primarily supported by small-scale non-agricultural industries have experienced rapid growth [13]. These regions capitalize on their inherent resource endowments and industrial advantages, attracting a significant influx of immigrants and capital, transforming into mixed-use villages that integrate living and production activities. They have developed distinct characteristics, such as the establishment of mature industrial chains encompassing industrial production, modern commerce, and leisure services [14]. Notably, these areas exhibit prominent features of a thriving private economy and distinctive local characteristics. The transformation of rural mixed-use functions has led to a significant influx of high-energy, high-pollution, and high-input industries into villages, resulting in a sharp increase in overall carbon emissions in rural regions [15]. Scholars have posited that mixed-use villages have become one of the main sources of China's carbon emissions [16], which could hinder China's efforts towards its "Dual Carbon" goals. At the same time, this has led to various social, economic, and environmental problems [17], such as harm to the rural environment, negative effects on villagers' health, and a widening gap between urban and rural areas, rendering rural development unsustainable.

However, there is a conspicuous absence of studies addressing the carbon emission characteristics and sustainable development concerns of mixed-use villages [16], representing a significant research gap in this field. Mainstream research posits that sustainability is shaped by three pillars: economy, environment, and society [18]. In this study, we define the sustainability of mixed-use villages as a balancing act between ecological environmental protection and industrial economic growth, and we aim to help achieve sustainable development and enhance the quality of rural living and working through the insights provided herein.

The remainder of this paper is structured as follows: In Section 2, we review the relevant publications in the literature on mixed-use villages. Section 3 explains the materials and methods employed for this study. Section 4 presents our research findings. In Section 5, we discuss the survey results and further propose sustainable development pathways for

lowering carbon emissions. Finally, in Section 6, we summarize the theoretical significance of this paper, identify future research directions, and acknowledge this study's limitations.

## 2. Literature Review

### 2.1. Mixed-Use Villages

Many researchers have engaged in academic research on mixed-use villages, and their research directions can be summarized as follows: (1) Regional growth perspective—Kong et al. proposed the concept of "working and living integration" as a driving force behind regional growth, providing empirical evidence through using productive rural settlements in the Yangtze River Delta region as examples [19]. (2) Economic development perspective—Triyuliana et al. analyzed the feedback effect of private individual's operational methods [20], such as family workshops, on rural economy and human settlement development in industrial villages. (3) Urban–Rural construction perspective—Chen et al. elucidated the current situation of mixed functions in villages [21], highlighting the synergistic effect of rural industrial development and urban–rural construction. (4) Spatial simulation perspective—Ma et al. simulated the spatial construction process of mixed-use villages and proposed a mechanism of spatial evolution under self-organizing effects [22]. Moreover, many researchers have employed various methods to assess the multifunctional vitality of urban and villages [23]. Jaroszewicz et al. utilized GIS and spatial data simulation to assess the vitality of mixed-use communities [24]. Hoppenbrouwer et al. proposed a method involving the use of spatial grammar and the mixed-use index (MXI) to assess the intensity of land use mix [25]. Zhu et al. proposed a multidimensional integrated approach to calculate mixed-use vitality and simulated the distribution characteristics of mixed-use spaces [26].

The above-mentioned studies provided important reference points for the development of this study. Existing research on mixed-use villages largely focuses on a macroscopic perspective, leaving gaps and methodological shortcomings in understanding micro-level endogenous dynamics and spatial organization. Therefore, targeted research exploration is urgently needed.

### 2.2. Rural Carbon Emissions

Research on carbon emissions in rural areas predominantly centers around emissions derived from agricultural operations and the rural living environment. The former encompasses energy consumption attributable to agricultural machinery and the application of chemical fertilizers and pesticides [27,28]. Studies have identified a u-shaped Environmental Kuznets Curve (EKC) relationship between agricultural production and emissions [29]. Conversely, carbon emissions from the rural living environment encompass energy consumption related to cooling, heating, transportation, and other daily life activities [30,31]. With the aging of the rural population, there has been a marked increase in carbon emissions resulting from residential energy consumption [32]. Scholars have observed that between 2012 and 2018, per capita carbon emissions in rural areas increased, even though overall emissions declined [33]. Presently, research strategies for reducing rural carbon emissions primarily focus on enhancing the carbon sequestration capabilities of rural ecosystems (such as forests, wetlands, and farmlands) and curbing carbon emissions from rural residential living. For instance, in-depth investigations have been conducted on the carbon balance and reserve fluctuations within forest ecosystems [34,35]. Additionally, studies have proposed strategies to amend rural consumption habits to reduce per capita carbon emissions [36].

Although the existing research in the literature is extensive, there remains a pronounced lack of studies addressing the equilibrium between enhancing the quality of work and life in rural areas and advancing energy conservation and emission reduction, especially in the context of the burgeoning emergence of mixed-use villages. Therefore, effectively identifying the characteristics of mixed-use villages in rural settings, establishing regionally tailored carbon assessment models, and exploring pathways to achieve sustain-

able development are the pivotal focal points of this research. Additionally, quantitative research on carbon emissions predominantly focuses on national, provincial, or major urban scales, meaning that studies that focus on smaller-scale areas such as villages are notably scarce. Indeed, narrowing the analysis to the micro level of individual households can reflect bottom-up endogenous patterns. Based on this, for the present study, we examined the relationship between the intensity of mixed-use villages, carbon emissions, and sustainability from a micro perspective.

## 3. Materials and Methods

### 3.1. Study Case

The study area is located in the Yangtze River Delta region, situated in eastern China's lower reaches of the Yangtze River. Encompassing an area of 358,000 km$^2$, the Yangtze River Delta region includes Shanghai, the Jiangsu Province, the Zhejiang Province, and the Anhui Province. As of the end of 2022, the region had a population of 237 million, making it one of the most densely populated areas in China [13,37]. According to administrative division data, in 2022, there were approximately 3100 villages in the Yangtze River Delta region, accounting for 9.4% of the total number in China [38].

As a pioneer in the implementation of rural revitalization strategies, the Yangtze River Delta region has undergone rapid socio-economic transformations, with industrial diversification emerging as a prominent feature [38]. This evolution has given rise to a variety of mixed-use rural typologies. Within these villages, family handicrafts, modern commerce, and tourism services have progressively supplanted traditional agriculture as the dominant industries. This shift has also fostered an integration of living and non-agricultural working activities within the rural milieu. Presently, mixed-use villages constitute no less than 60% of the total rural communities in the Yangtze River Delta region [39], marking the highest proportion nationwide. However, the proliferation of mixed-use villages has precipitated a notable surge in per capita carbon emissions in the region, which have risen by 3.56% over the past decade. Additionally, energy consumption in rural areas accounts for approximately 40% of the total, with carbon emissions surpassing 45% of the total [40]. Under the "Dual Carbon" goals, achieving a balance between developing the sustainability of the economy and the environment in the Yangtze River Delta region becomes increasingly crucial. Simultaneously, as a demonstration zone for integrated ecological and green development, the Yangtze River Delta region bears the responsibility of setting an exemplary precedent and providing incentives for low-carbon transitions in other regions [41]. Hence, the mixed-use villages of the Yangtze River Delta region were selected as the areas of focus for this research study.

Initially, we consulted official statistical yearbooks, the China Rural Development Report (2021) [42], and the China Rural Revitalization Development Report (2021) [43] to compare basic information such as the industrial structure, population size, regional area, and geographical location of rural areas, preliminarily selecting 150 potential cases (accounting for approximately 5% of the total). Following a six-month period of on-site investigation and data collection by our team (from June to December 2021) and discussions with several scholars, eight research cases were eventually identified (representing about 5% of the potential cases). These eight villages met the following criteria: (1) They each have distinct dominant industries and display noticeable differences in mixed-use characteristics. (2) The population sizes, regional areas, and scales of work and living are comparable across each village. (3) Metrics like mixed-use intensity and carbon emissions are readily accessible. (4) These villages are distributed across the following regions of the Yangtze River Delta: Zhejiang (3 villages), Jiangsu (2 villages), Anhui (2 villages), and Shanghai (1 village) (Figure 1).

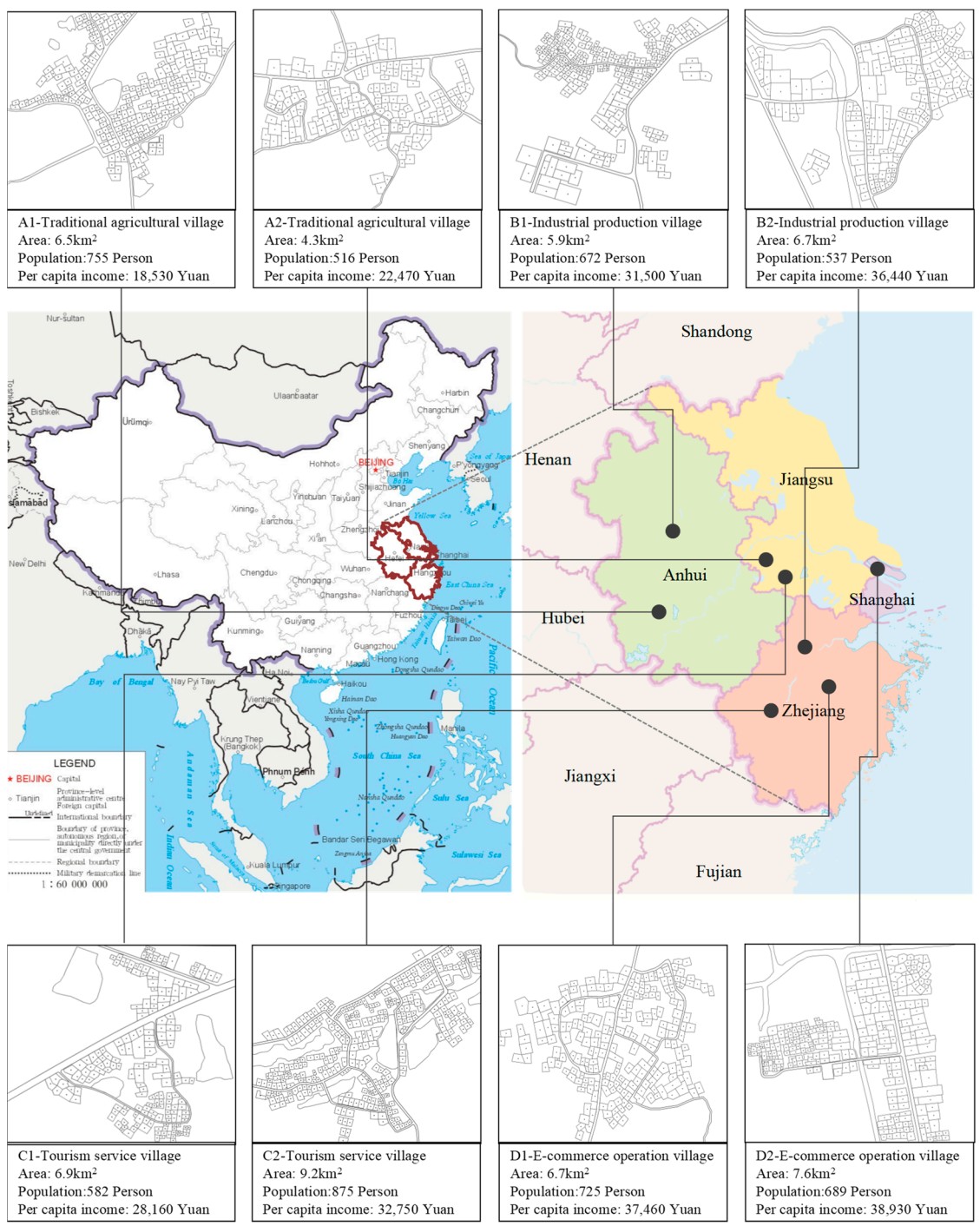

**Figure 1.** Basic information regarding the eight selected mixed-use villages. (The cartographic data were derived from the Chinese Standard Map Service System, while the rural information was obtained through remote sensing technology and field surveys.)

Based on studies by scholars [44], rural development types in the eastern coastal region of China are diverse and broadly categorized into six dominant types: agriculture-led, fishery-centric, industry-focused, trade-driven, tourism-oriented, and comprehensive development. In this study, based on a preliminary survey analysis, we adjusted and consolidated these six types, eventually identifying four industry-leading rural types. They are as follows: traditional agricultural villages (A1 and A2), industrial production villages (B1 and B2), tourism service villages (C1 and C2), and E-commerce operation villages (D1 and D2). These four categories are mutually exclusive and collectively encompass over 95%

of the mixed-use rural types in the Yangtze River Delta. Based on initial data, these village types account for 28%, 25%, 22%, and 22% of the mixed-use villages in the region, respectively. Traditional agricultural villages mainly derive income from agricultural production. Industrial production villages predominantly focus on manufacturing, typically hosting large factories or enterprises. Tourism service villages stimulate rural economic development through tourism, involving businesses like homestays, dining establishments, and retail businesses. E-commerce operation villages primarily sell products online, possessing significant warehouses and logistics spaces within the village.

### 3.2. Data Collection

This study draws on data from two main sources: statistical data and empirical investigations [45]. The statistical data used were primarily obtained from the Rural Energy Yearbook (2022) [46], Environmental Quality Reports (2021) [47], and online statistical databases. Empirical investigations were collected through departmental visits and on-site investigations. The departmental visits involved gathering data from the administrative bodies overseeing the villages. The on-site investigations encompassed three aspects: (1) behavioral and cognitive surveys recorded villagers' daily working and living activities using questionnaires and interviews, (2) spatial form surveys aimed to define the boundaries of rural carbon emission units, (3) energy consumption surveys combined questionnaire surveys and visits to administrative bodies to obtain relevant energy consumption data.

During the period from June 2021 to December 2021, we conducted our investigation using on-site surveys, semi-structured interviews, and distributed questionnaires. Additionally, in March 2022, in-depth interviews were conducted to obtain supplementary data. The questionnaires and interviews primarily focused on topics such as the developmental history of rural industries, changes in employment patterns, the degree of integration between production and residence, and electricity and energy consumption. The interviewees represented various entities, including rural enterprises, individual businesses, village officials, skilled individuals, and ordinary villagers. The data and information described above form the basis of this study.

### 3.3. Defining the Research Boundaries

This study concentrates on micro-scaled mixed-use units as the basis for defining research boundaries. Mixed-use units can be conceptualized as a single building or a contiguous cluster of buildings [16], encapsulating village residential zones, factories, markets, and other explicitly defined spatial entities. Such units typically exhibit distinct spatial demarcations in the form of walls, courtyards, and building exteriors. However, field investigations reveal a prevalent trend in rural settings where residents neither confine their work nor their lives strictly within these demarcated zones but rather encroach upon semi-public spaces surrounding the units [48]. By specifically identifying the functions of these semi-public spaces on site, we supplemented and adjusted the actual boundaries of the units (Figure 2a). The Spatial Statistics Grid Method (STING) [49] was employed to construct mixed-use units for explaining the distribution of mixed-use intensity and carbon emissions in the four types of villages. In this context, each village can be subdivided into multiple mixed-use units (Figure 2b). Due to land restrictions in rural residences, barring large markets and factories, the majority of these units generally span an area ranging from 80 to 140 m$^2$.

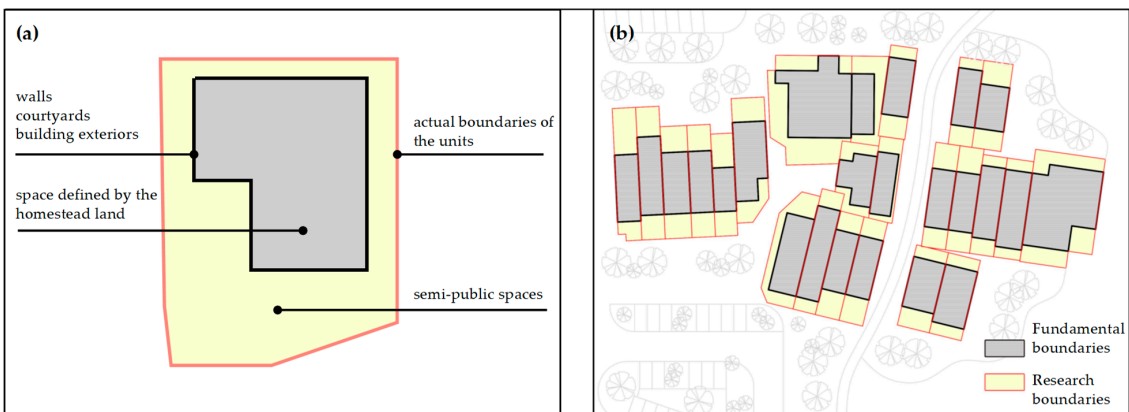

**Figure 2.** Defining the research boundaries and constructing mixed-use units.

*3.4. Calculation and Data Analysis*

3.4.1. Mixed-Use Intensity

The mixed-use intensity in villages needs to be considered from multiple dimensions [16], including the following: (1) Temporal Dimension (*Mix T*): the duration ratio of working and living activities, reflecting the temporal characteristics of mixed-use functionality in rural areas. (2) Spatial Dimension (*Mix S*): the ratio of working and living areas, indicating the functional composition of rural mixed-use units. (3) Population Dimension (*Mix P*): the ratio of working and living populations in mixed-use villages.

$$Mix\ T = [1 - max(t_w, t_l)]/(t_w + t_l) \times 2 \tag{1}$$

$$Mix\ S = [1 - max(s_w, s_l)]/(s_w + s_l + s_a) \times 2 \tag{2}$$

$$Mix\ P = [1 - max(p_w, p_l)]/(p_w + p_l + p_t) \times 2 \tag{3}$$

$$Mix = \sqrt[3]{\alpha Mix\ T \times \beta Mix\ S \times \gamma Mix\ P} \tag{4}$$

where $t_w$ and $t_l$ represent the duration of working and living activities, respectively, and $max(t_w, t_l)$ indicates the larger value between the two. When $t_w = t_l$, *Mix T* reaches its maximum value of 1. In the spatial dimension, $s_w$ represents the working area, $s_l$ represents the living area, and $s_a$ represents the auxiliary area. In the social dimension, $p_w$ represents the working population, $p_l$ represents the living population, and $p_t$ represents temporary personnel. $\alpha$, $\beta$, and $\gamma$ are weight values for the temporal, spatial, and population dimensions, respectively. To ensure the balanced influence on the mixed-use status of villages, we set $\alpha = \beta = \gamma = 1$. Based on the final calculated values, the range of *Mix* is set from 0 to 1. A higher *Mix* value indicates a higher level of mixed-use intensity in the rural areas, while a lower value indicates a lower level of mixed-use intensity.

3.4.2. Carbon Emissions

In rural areas, energy input, transfer, and output are ultimately reflected in the final composition of carbon emissions. This encompasses carbon emissions of both living and working energy consumption, as well as carbon emissions of transportation within and between villages [50]. Given that $CO_2$ is the primary greenhouse gas and that data related to are more readily available, this study primarily focuses on carbon emissions dominated by $CO_2$.

(1)  Direct carbon emissions of fuels for working and living:

$$E_h = \sum_{i=0}^{n} \varepsilon_i \cdot E_i \tag{5}$$

where $E_h$ is the carbon emissions of fuel, $i$ represents the $i$-th type of energy resource, $\varepsilon_i$ is the carbon emission coefficient of the $i$-th energy source (Table 1). The data were extracted from IPCC guidelines for national greenhouse gas inventories (2006) [51]; $E_i$ is the amount of fuel used, and $n$ is the number of energy types.

(2) Indirect carbon emissions of electric power for working and living:

$$E_e = E_f \cdot \gamma \tag{6}$$

where $E_e$ is the carbon emissions of electric power, $E_f$ represents the electric power consumption obtained from China Rural Energy Yearbook (2022) [46], which were released by the local power companies. $\gamma$ is the electricity emission factor, derived from the Baseline Emission Factors for Regional Power Grids in China [17], with a value of 0.928 kg $(CO_2)$/kWh.

(3) Carbon emissions of transportation:

$$E_w = \sum_{i,j}^{n} \left( C_{i,j} \cdot D_{i,j} \cdot P_{i,j} \right) \tag{7}$$

where $E_w$ represents the total carbon emissions calculated based on the vehicle's driving distance. $i$ represents the vehicle type (e.g., cars, motorcycles), and $j$ represents the fuel type (e.g., gasoline, diesel). $C_{i,j}$ represents the number of vehicles, $D_{i,j}$ denotes the annual kilometers driven per vehicle type (km), and $P_{i,j}$ is the average carbon emissions per kilometer for each vehicle type (kg/km).

**Table 1.** Carbon emission coefficient for rural fuels.

| Coding | Coefficient Name | Coefficient Value | Coding | Coefficient Name | Coefficient Value |
|---|---|---|---|---|---|
| $\varepsilon_1$ | Coal | 2.689 t $(CO_2)$/tce | $\varepsilon_5$ | Coking coal | 0.414 t $(CO_2)$/tce |
| $\varepsilon_2$ | Gasoline | 2.027 t $(CO_2)$/tce | $\varepsilon_6$ | Straw | 1.247 t $(CO_2)$/t |
| $\varepsilon_3$ | Diesel | 2.166 t $(CO_2)$/tce | $\varepsilon_7$ | Firewood | 1.436 t $(CO_2)$/t |
| $\varepsilon_4$ | Natural gas | 1.624 t $(CO_2)$/tce | $\varepsilon_8$ | Biogas | 11.720 t $(CO_2)/10^4$ m$^3$ |

The total carbon emissions calculation in mixed-use villages is the sum of the direct carbon emissions of fuels ($E_h$), indirect carbon emissions of electric power ($E_e$), and transportation-related carbon emissions ($E_w$). The calculation formula is as follows:

$$E = E_h + E_e + E_w \tag{8}$$

*3.5. Spatial–Temporal Characteristics Analysis*

(1) Calculating the mixed-use intensity and carbon emissions of each unit. Using the calculation method in Equations (1)~(4), we determined the mixed-use intensity of each unit and established a database for rural mixed-use intensity. Similarly, the carbon emissions of each unit were computed utilizing Equations (5)~(8), subsequently leading to the establishment of a dedicated database.

(2) Data classification. To amplify the visualization of spatial distribution and precisely depict the spatio-temporal variations in the mixed-use intensity and carbon emissions of a given village, we adopted the mean-standard deviation method for classification based on the computed results [52]. This approach effectively illustrates the concentration and dispersion characteries of data. The detailed classification criteria are presented in Tables 2 and 3.

(3) Spatial characteristics analysis. Drawing upon the classification of mixed-use intensity and carbon emissions data, different levels of shading were applied to represent individual mixed-use units. This provided a thorough understanding of the spatial distribution patterns of mixed-use intensity and carbon emissions within the village [53].

Subsequently, a correlation analysis of mixed-use intensity and carbon emissions was conducted, as detailed in the following equation.

$$C = (N_1 + 0.5N_2)/N_T \qquad (9)$$

where $C$ represents the correlation value, $N_1$ denotes the number of units with identical classification levels for mixed-use intensity and carbon emissions, $N_2$ signifies the number of units with adjacent classification levels for mixed-use intensity and carbon emissions, and $N_T$ stands for the total number of units within the mixed-use village.

**Table 2.** Data classification regarding carbon emissions.

| Data | Classification Level | Classification Method |
|---|---|---|
| Carbon emissions | Low carbon emissions | [min, mean-standard deviation] |
| | Low–medium carbon emissions | (mean-standard deviation, mean-0.5 × standard deviation] |
| | Medium carbon emissions | (mean-0.5 × standard deviation, mean+0.5 × standard deviation] |
| | Medium–high carbon emissions | (mean+0.5 × standard deviation, mean × standard deviation] |
| | High carbon emissions | (mean+standard deviation, max] |

**Table 3.** Data classification regarding mixed-use intensity.

| Data | Classification Level | Classification Method |
|---|---|---|
| Mixed-use intensity | Low mixed-use intensity | [min, mean-standard deviation] |
| | Low–medium mixed-use intensity | (mean-standard deviation, mean-0.5 × standard deviation] |
| | Medium mixed-use intensity | (mean-0.5 × standard deviation, mean+0.5 × standard deviation] |
| | Medium–high mixed-use intensity | (mean+0.5 × standard deviation, mean + standard deviation] |
| | High carbon mixed-use intensity | (mean+standard deviation, max] |

Utilizing GIS tools, we performed a spatial heatmap visualization analysis [54] of mixed-use intensity and carbon emissions data to depict the spatial variations in the entire rural area. This approach effectively illustrates the spatial changes in mixed-use intensity and carbon emissions across the rural areas.

(4) Temporal characteristics analysis. Based on the method described in step (3), we identified the features of mixed-use intensity and carbon emissions for mixed-use villages at different time points. By comparing these features across various times, we can obtain a clear view of the regular changes in mixed-use intensity and carbon emissions. This approach helps us see details that might be hard to spot when looking at space features at just one time point (Figure 3).

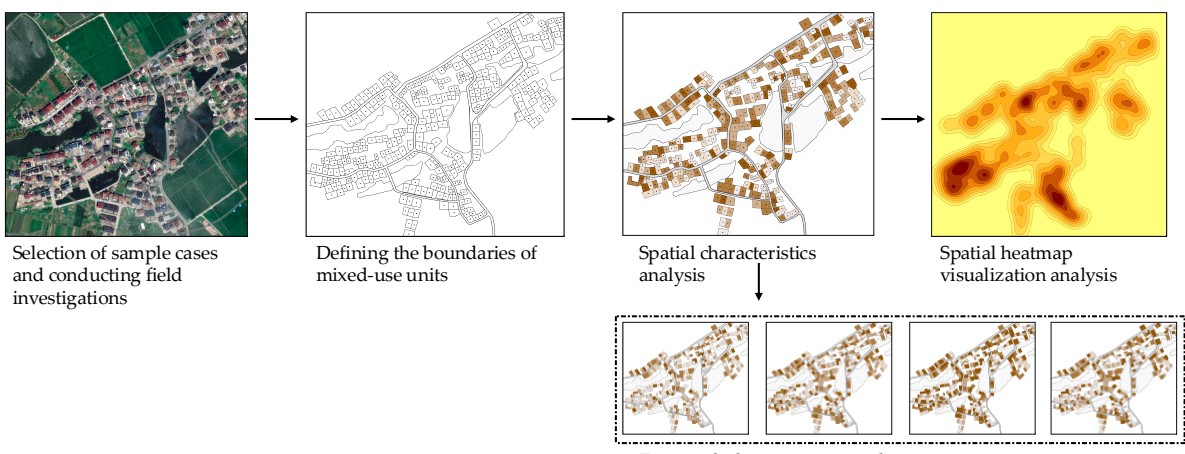

Selection of sample cases and conducting field investigations

Defining the boundaries of mixed-use units

Spatial characteristics analysis

Spatial heatmap visualization analysis

Temporal characteristics analysis

**Figure 3.** Process visualization of the spatial–temporal characteristics analysis.

## 4. Results

### 4.1. Mixed-Use Intensity and Carbon Emissions of Four Types of Villages

The results (Figure 4) show that in traditional agricultural villages, almost every household is engaged in agricultural activities, resulting in similar industrial patterns with minimal variation influenced by location factors. As a result, the mixed-use intensity between units is relatively small. In this type of village, the majority of units exhibit low and low–medium mixed-use intensity (A1-28% and 30%, A2-25% and 29%). The main reason for this is the limited number of employed individuals, shorter working hours, smaller industrial areas, and a predominant focus on residential activities within each unit. This type of village has the highest number of low carbon emission units (A1-31%, A2-27%). This can be attributed to the absence of large-scale production and processing factories within the villages, with carbon emissions mainly originating from rudimentary agricultural processing, transportation, and storage activities. From Table 4, it is evident that there is no apparent correlation (A1-0.338, A2-0.356) between mixed-use intensity and carbon emissions in traditional agricultural villages.

**Table 4.** Evaluation of mixed-use intensity and carbon emissions of eight villages.

| Type | Mixed-Use Intensity (Most Dominant) | Carbon Emissions (Most Dominant) | Correlation |
|---|---|---|---|
| A1—Traditional agricultural | Low-medium | Low | 0.338 |
| A2—Traditional agricultural | Low-medium | Low-medium | 0.356 |
| B1—Industrial production | Medium | Medium–high | 0.578 |
| B2—Industrial production | Medium–high | Medium–high | 0.632 |
| C1—Tourism service | Medium | Medium | 0.495 |
| C2—Tourism service | Medium | Medium–high | 0.451 |
| D1—E-commerce operation | Medium–high | Medium | 0.674 |
| D2—E-commerce operation | Medium–high | Medium | 0.653 |

In the industrial production villages, the majority of units exhibit medium and medium–high mixed-use intensity (B1-23% and 22%; B2-21% and 27%). Furthermore, the fluctuation in mixed-use intensity among these units is relatively minimal, which can be attributed to the fact that almost all households engage in related processing industries or their derivatives to varying degrees. The majority of high-carbon-emitting units in villages are factories or family workshops, as the machinery involved in production and processing requires high energy consumption. The majority of units exhibit medium–high and high carbon emissions (B1-27% and 18%, B2-25% and 20%), contributing to the overall highest carbon emissions in the villages. There is a strong correlation (B1-0.578, B2-0.632) between high mixed-use intensity units and high carbon emissions units in villages, with a distribution pattern centered around large factories and gradually decreasing towards distant areas.

In the tourism service villages, approximately half of the units have mixed-use intensity values that fall between low–medium and medium levels. Due to the seasonal nature of the tourism industry and the initial economic investments required for tourism services (e.g., transforming houses into guesthouses), these rural areas tend to exhibit lower overall mixed-use intensity values. However, there is a clear spatial tendency in the distribution, with higher mixed-use intensity units concentrated along well-connected roads or near scenic centers. Within the tourism service villages, the number of units with low–medium, medium, and medium–high carbon emissions are quite similar (C1-22%, 26%, and 24%, C2-25%, 21%, and 25%). The main sources of carbon emissions in these units are related to transportation, cooking, refrigeration, heating, and lighting, resulting in relatively minor differences among them. Overall, the carbon emissions in tourism service villages are relatively low, and there exists a certain correlation (C1-0.495, C2-0.451) between carbon emissions and mixed-use intensity for the units.

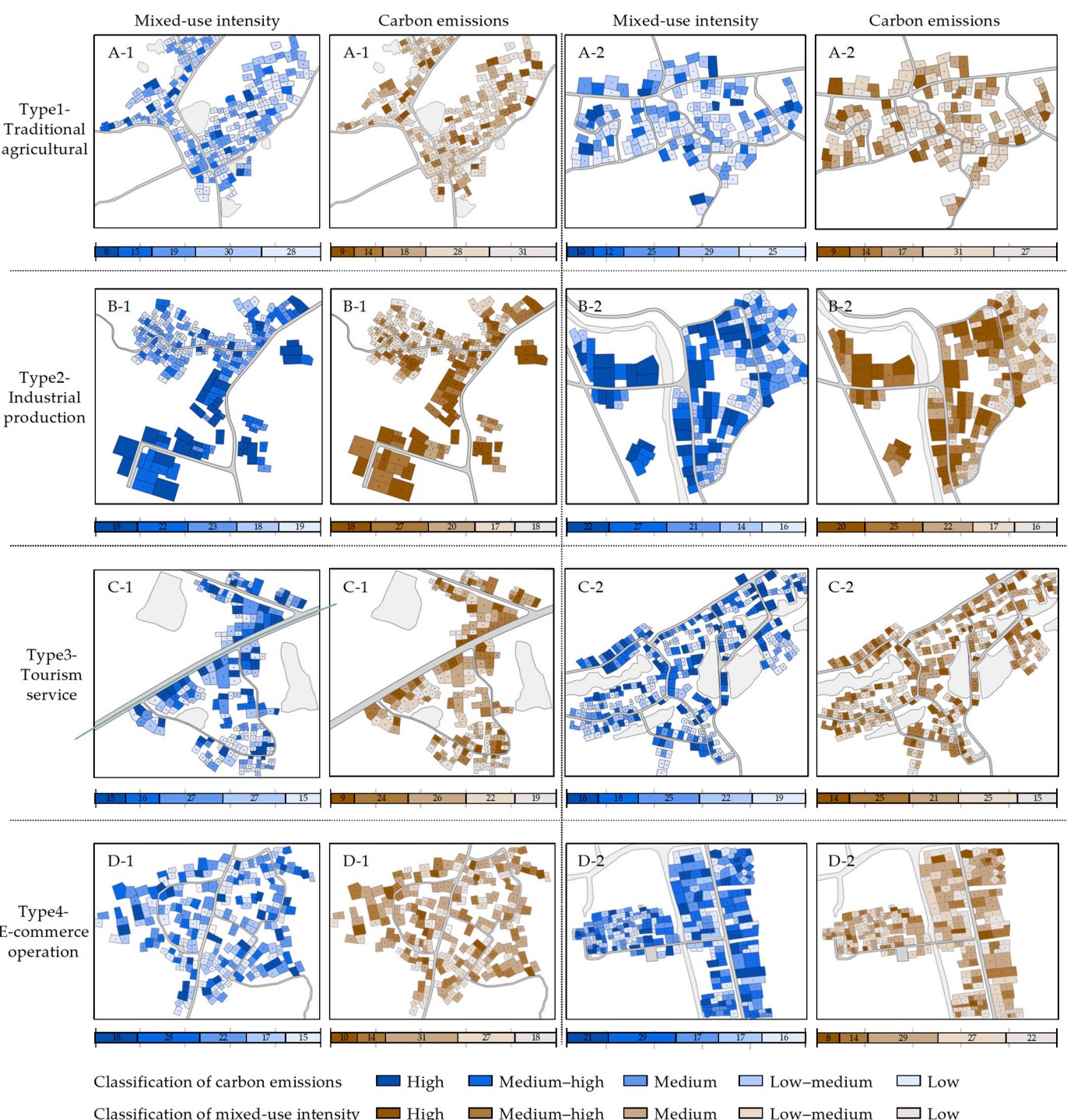

**Figure 4.** The mixed-use intensity and carbon emission patterns in typical villages of the Yangtze River Delta region. (Each village has two corresponding analysis results: mixed-use intensity and carbon emission. The numbers in the bars represent quantitative proportions of units.)

In E-commerce operation villages, the proportion of units with medium–high mixed-use intensity is the highest (D1-28% and D2-29%), resulting in the overall highest mixed-use intensity among the four types, which is related to the tendency of mutual imitation and easy dissemination in E-commerce operations. In this type of village, units with higher mixed-use intensity values are distributed without a clear patter, both along the roadsides and within the villages, showing relatively insignificant variations in mixed-use intensity. E-commerce operation villages have the most units with medium carbon emissions (D1-

31% and D2-29%). The main sources of carbon emissions are market operations, logistics transportation, or packaging processing. The carbon emissions of units in these villages are closely related to the number of employees and working hours, showing a significant correlation (D1-0.674, D2-0.653) with mixed-use intensity.

### 4.2. Spatial Characteristics

Using GIS, we conducted a spatial typology analysis of the carbon emissions in the four types of villages. The results (Figure 5) are as follows:

(1) The carbon spatial tendency of traditional agricultural villages is relatively weak, with no obvious high carbon emission areas. In this type of rural area, units with relatively higher carbon emissions are scattered throughout the village, which is due to the weak correlation between agricultural production and geographical location. From A1 and A2 villages, it can be observed that the high carbon emission concentration area is small and exhibits a random distribution pattern, resulting in the overall lowest carbon emissions.

(2) The carbon spatial tendency of industrial production villages exhibit significant tendencies, always manifesting as clustered block-like formations in high-carbon regions. In this type of rural area, carbon-emitting units are interlinked in industrial chains, and some units spontaneously form "production alliances", integrating manufacturing, processing, and transportation functions. Consequently, the industrial model exhibits replicative expansion within these clusters, leading to distinct regional tendencies. In B1 and B2 villages, high-carbon regions are observed near large factories or in close proximity to family workshops with significant land areas. Although each segment of the industrial chain may have different carbon emissions, such as higher emissions during initial coarse processing and lower emissions during post-production fine processing, the overall carbon core characteristic remains the most pronounced.

(3) The carbon spatial tendency of tourism service villages is evident. Due to the pronounced demand for clientele, units exhibit significant agglomeration effects, primarily located along convenient transportation routes or in scenic areas that attract more tourists, resulting in higher carbon emissions in these regions. Conversely, units with limited accessibility and fewer natural resources have lower carbon emissions as they sporadically receive tourists during peak tourism seasons. Analyzing C1 and C2 villages, the carbon emissions show a decreasing gradient from along the streets towards the interior of the villages, indicating noticeable differences between the outer and inner regions.

(4) The carbon spatial tendency of E-commerce operation villages is generally moderate, with high carbon emission areas mainly concentrated near large markets or main roads. A significant number of operating units are distributed along the main streets or on both sides of large markets, providing convenience for procurement and logistics. However, there are also many operating units located away from roads and markets, attracted by relatively lower rent and larger building spaces. Nevertheless, in D1 and D2 villages, the core areas with high carbon emissions are relatively scarce due to the nature of E-commerce operations, which do not generate excessive carbon emissions.

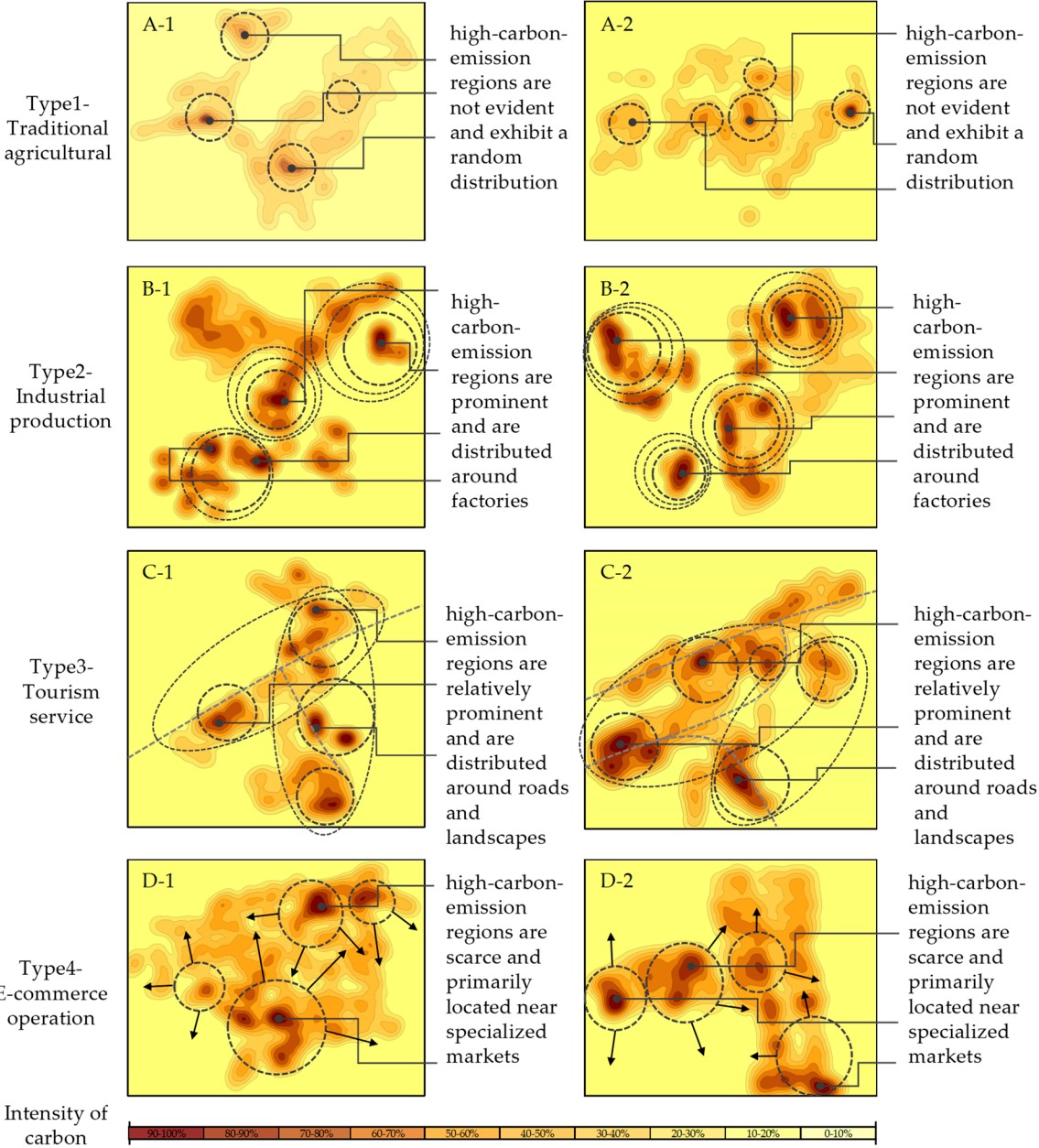

**Figure 5.** GIS mapping of the spatial distribution of carbon emissions from mixed-use units (Each figure corresponds to the analysis result of a village).

### 4.3. Temporal Characteristics

The mixed-use characteristics of villages determine the periodic variation differences in energy consumption and carbon emissions, as well as the volatility of these changes.

(1) Traditional agricultural villages exhibit low volatility. The primary carbon sources in these villages are related to residential energy use and agricultural activities. The peak periods of carbon emissions in agriculture and forestry are mainly concentrated in spring and autumn (sowing and harvesting seasons), while the low periods are concentrated in summer and winter. For residential energy use, the carbon emissions peak in the summer and winter seasons (Figure 6). The peak periods of carbon emissions for production and living overlap only briefly, resulting in a relatively stable overall carbon emission pattern with fluctuations within 30% of the total.

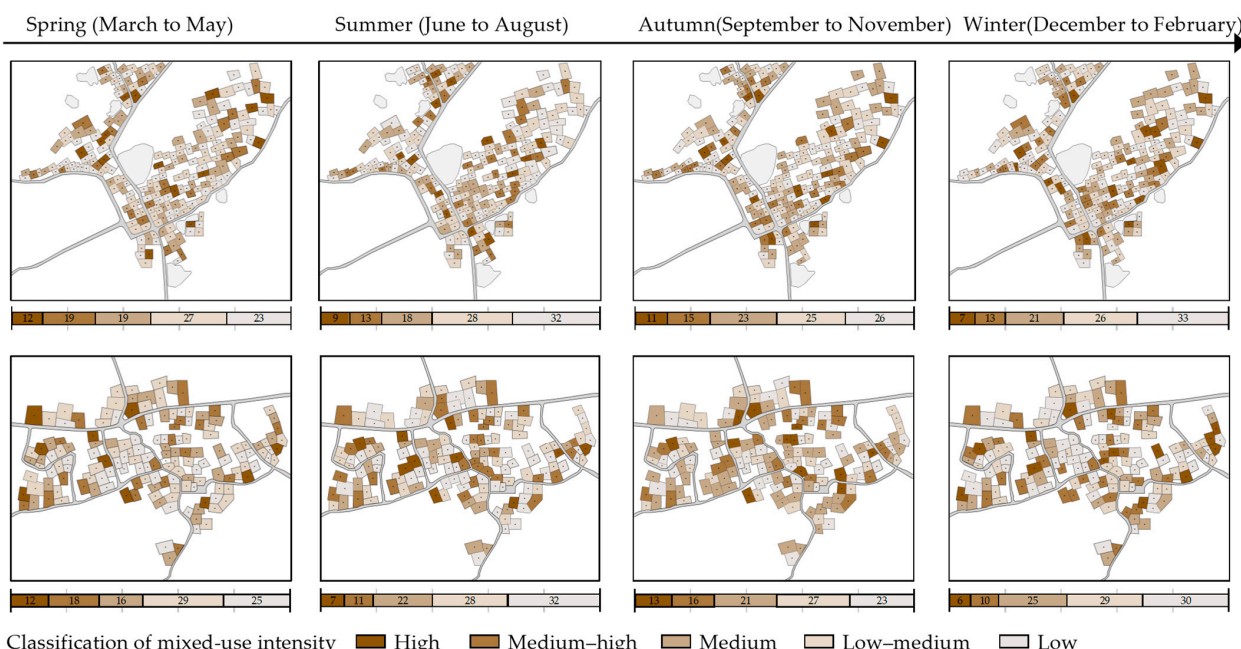

**Figure 6.** The spatial periodic variation of carbon emissions in traditional agricultural villages.

(2) Industrial production villages exhibit significant fluctuations in carbon emissions. The carbon sources in these villages mainly consist of industrial production, transportation energy, and household energy consumption. Due to the configuration of the industrial supply chain, the production distribution in these villages follows a semi-fixed pattern. In response to changes in product demand and order rhythms, production intensity experiences periodic variations. Carbon emission peaks typically concentrate in the summer (peak production season), while lows occur in the winter (off-peak production season). The fluctuation range remains within 60% of the total emissions (Figure 7).

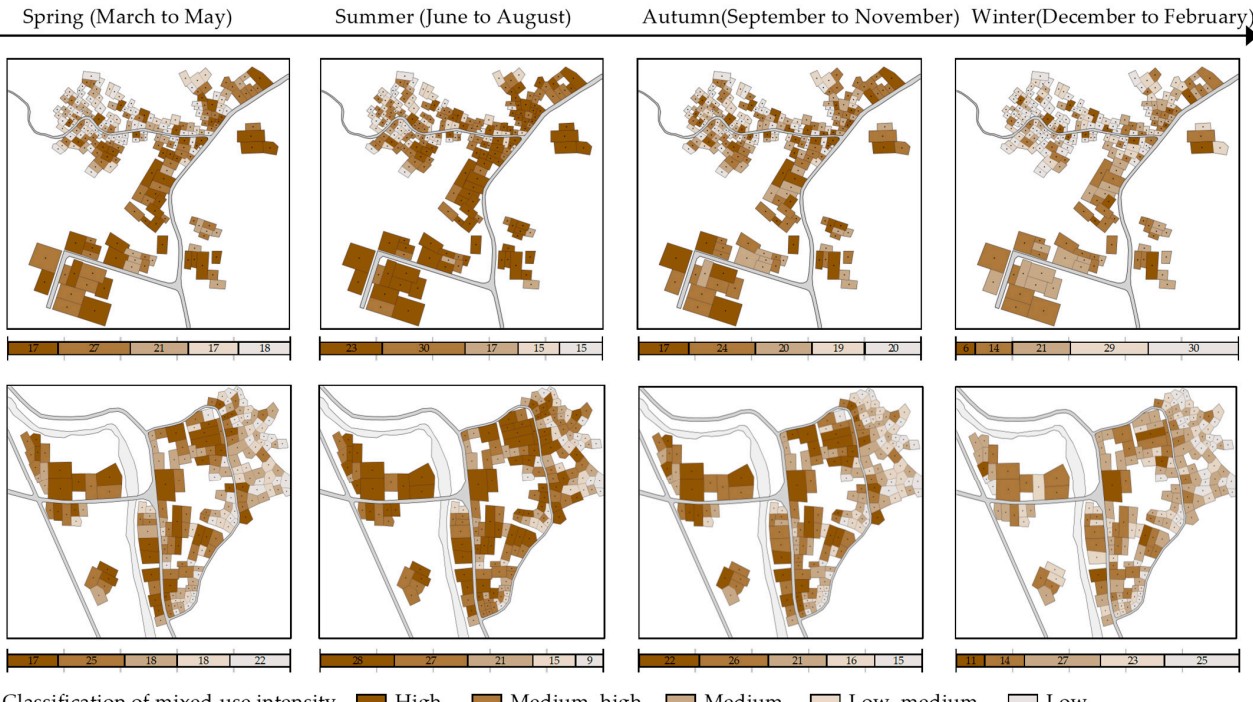

**Figure 7.** The spatial periodic variation of carbon emissions in industrial production villages.

(3) Tourism service villages exhibit the highest fluctuations in carbon emissions. The carbon emissions in these villages mainly derive from the service industry and transportation. Due to the significant influence of festivals and seasons on leisure tourism industry, the carbon emissions show the most evident fluctuations, even exceeding 150% fluctuation, surpassing the environmental tolerance threshold. Peak emissions are concentrated in February (Chinese New Year) and from May to October (summer vacation and various holidays) (Figure 8). Especially during the tourism peak weeks with rising accommodation prices, even households that were previously purely residential may temporarily engage in tourism services, leading to a substantial increase in carbon emissions.

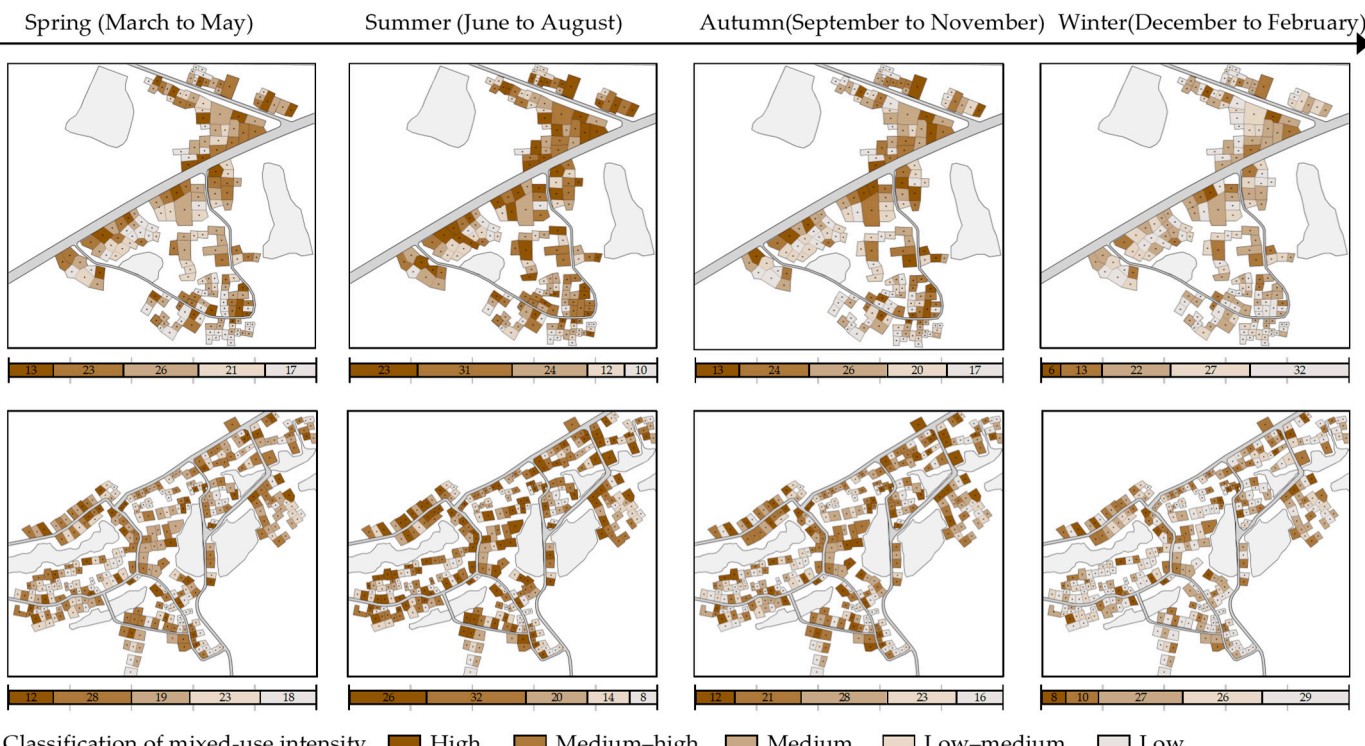

**Figure 8.** The spatial periodic variation of carbon emissions in tourism service villages.

(4) E-commerce operation villages exhibit the smallest fluctuations in carbon emissions. The carbon emissions in these villages mainly derive from office operations, warehousing, and transportation. The carbon emissions in this type of village show little periodic variation throughout the year, with minimal differences in production and operations across seasons (Figure 9). However, during large promotional events and festivals, there could be sharp short-term increases in carbon emissions, with fluctuation amplitudes ranging from 50% to 100%. Overall, these villages demonstrate high stability in carbon emissions and are minimally affected by climate and seasonal factors.

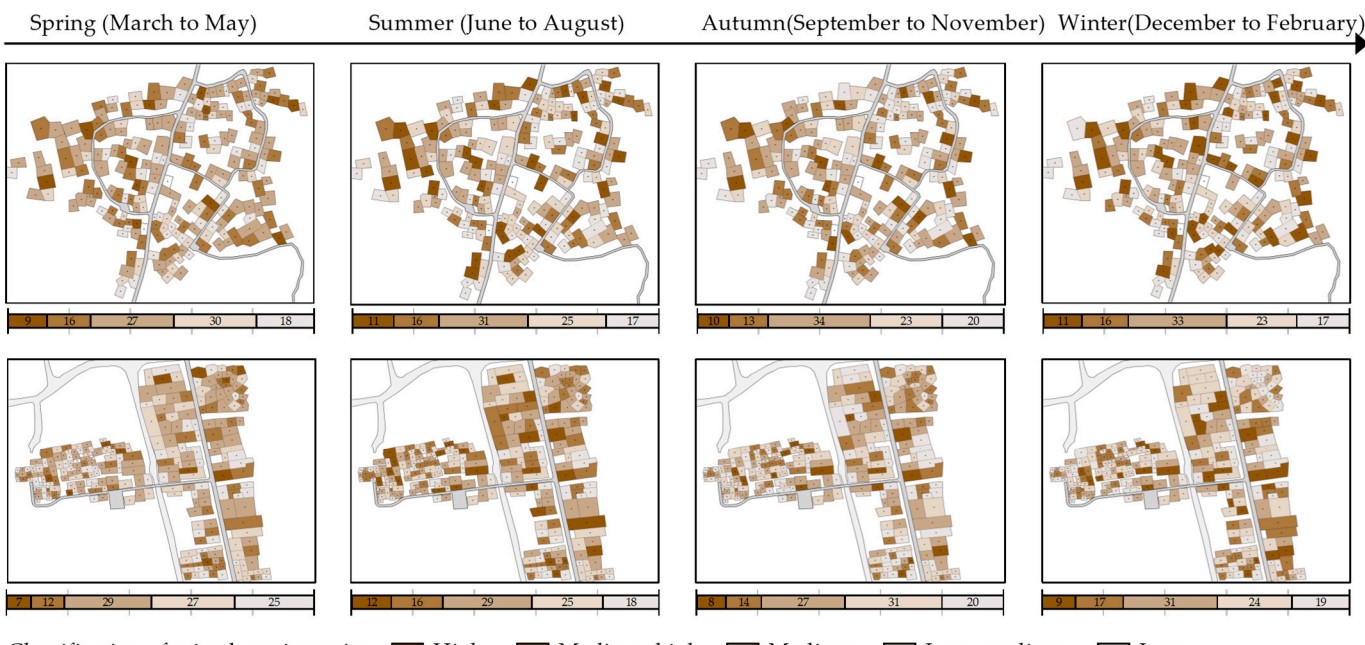

**Figure 9.** The spatial periodic variation of carbon emissions in E-commerce operation villages.

## 5. Discussion

### 5.1. Interpretation of Findings

In comparing the four villages, distinct patterns in mixed-use intensity emerged. Traditional agricultural villages predominantly show low to low–medium mixed-use intensity due to their limited industrial scope and predominant residential activities [55]. Contrastingly, industrial production villages exhibit medium to medium–high mixed-use intensity due to their inhabitants' widespread engagement in processing industries [56]. Tourism service villages present a varied intensity that is influenced by the seasonal nature of tourism and initial economic investments required [57]. E-commerce operation villages demonstrate the highest mixed-use intensity, possibly due to the mutual imitation prevalent in E-commerce endeavors [58]. In essence, the core economic activities and infrastructural factors within each village significantly influence their mixed-use intensity.

The four types of mixed-use villages exhibit pronounced spatial heterogeneity in carbon emissions, largely attributed to the nature of their working activities. Traditional agricultural villages showcase a dispersed carbon emission pattern, aligning with the observations of Liang et al. that geographical location bears a weak influence on agricultural carbon emissions [59]. In contrast, the pronounced block-like carbon clusters in industrial production villages reflect the interconnectedness of mixed-use units, as seen in Xiao et al. [60]. Tourism service villages manifest the highest emission fluctuations, underscoring the volatile nature of the tourism industry and its susceptibility to seasonality and events [61]. Interestingly, E-commerce operation villages, despite being a relatively new typology, illustrate moderate spatial carbon emission tendencies and minimal seasonal fluctuations, reflecting the largely stable operational pattern of E-commerce, save for promotional events [62]. Essentially, the spatial and temporal characteristics of carbon emissions across various villages offer a foundational basis for policy formulation in an academic context.

As rural development globally continues to underscore the importance of sustainability, understanding the dynamics of mixed-use intensity and carbon emissions becomes paramount. Traditional agricultural villages, with their low carbon footprint, may indicate pathways toward sustainability; however, they might be economically vulnerable. In contrast, industrial villages might exhibit economic robustness but pose significant environmental challenges. E-commerce and tourism villages represent an intermediary

realm, underscoring the imperative of balancing growth. Thus, achieving sustainability necessitates tailored strategies for each village typology, integrating both economic growth and energy-saving carbon reduction dimensions.

*5.2. Implications*

The mixed-use intensity and carbon emissions in villages are inherently complex features, and there are challenges in their development and organization. Addressing the current contradictions between rural living, working, and environmental improvement requires more than just relying on formal data modeling analysis. It also necessitates avoiding the application of generalized mixed-use patterns based on experience. Instead, the key lies in adopting specific and detailed policy responses to effectively tackle these issues.

(1) Eliminating "carbon islands" through precise and targeted governance: Rural mixed-use units tend to aggregate around specific regions, leading to significantly higher carbon emissions in local areas than in peripheral regions, creating spatial "carbon islands" [63]. Taking industrial production villages (B1 and B2) as an example, within a range of 200–300 m around large factories, units exhibit high production inputs, a large workforce, and increased energy consumption and carbon emissions, resulting in pronounced "carbon island" areas. The key to eliminating "carbon islands" lies in the targeted governance of high-carbon-emitting units. Firstly, incentivizing policy measures is a key notion to consider. Diverse incentive and penalty mechanisms are provided in land, finance, fees, equipment, and technology, encouraging units to voluntarily transition to low-carbon operations, fostering regional synergies through imitation-based diffusion. Secondly, implementing standardized interventions by setting carbon emission targets for units, particularly mandatory regulations on carbon critical value, carbon production, carbon quotas, etc., [64] to prevent excessively high carbon emissions within specific times or spaces.

(2) Guiding the balance of "carbon flow" through ecological corridors for regulation: Road pathways, greenbelt corridors, and watercourse channels serve as conduits within the rural spatial structure [65]. Their connectivity can influence the penetration efficiency of external environmental resources, subsequently altering the aggregation order and manner of units. Taking tourism service villages (C1 and C2) as an example, the absence of a systematic corridor network within the village, coupled with poor internal–external connections and suboptimal diversion efficiency, results in a disproportionate tilt of a significant volume of public resources being allocated towards a minority of units. Adopting a "comb-like" or "grid-like" corridor network to connect clusters of units and subdividing land parcels into smaller plots (typically 0.2~0.5 hectares) to achieve seamless integration with rural public resources can enhance the accessibility of units within clusters. This encourages visitors to penetrate into the rural interior, guiding the balanced flow of carbon elements. Moreover, using landscape corridors as linkages to interconnect surrounding parks, green spaces, forests, waterways, and units can improve the living environment in rural areas, facilitating the carbon sequestration function of the rural green system [66].

(3) Inhibiting the growth of "carbon entropy" through the organic integration of units: Dispersed and independently operated mixed-use units are often individual households lacking effective shared platforms for land, labor, capital, equipment, management, and information, leading to inefficient resource utilization and disorganization. In E-commerce villages (D1 and D2), operating units lack a clear spatial aggregation pattern, being randomly scattered throughout the villages, resulting in a scattered distribution of carbon space and relatively high entropy. To reduce the spatial entropy of carbon emissions, external intervention and intervention are necessary to counteract internal dissipation tendencies. Decision makers can employ top-down land use zoning and facility integration to encourage and guide the integration of small operating units (preferably 4~8 households), merging and scaling production and operational elements moderately [67]. This approach promotes the formation of

efficient and synergistic mixed-use clusters, thereby guiding the orderly development of carbon emission spatial patterns. Within the mixed-use clusters, shared major nodes such as parking lots, warehouses, and packaging facilities can be established to facilitate integrated logistics and pedestrian flow while reducing unnecessary spatial and equipment energy waste [68].

This study offers actionable policy implications for the sustainable development of specific mixed-use villages in the Yangtze River Delta region, and these strategies can be applied to the majority of villages with similar characteristics. However, the types of mixed-use villages vary significantly across China and even globally, often leading to increased complexity in mixed characteristics and carbon emissions. Strategies such as eliminating "carbon islands", guiding the balance of "carbon flow", and inhibiting the growth of "carbon entropy" can provide valuable insights for other regions. Nevertheless, the actual planning or construction strategies should be adjusted and formulated based on the specific characteristics of individual villages at particular times and locations.

## 6. Conclusions

Diverging from prior studies, we adopted a bottom-up approach, leveraging the micro-level perspective of mixed-use units to quantitatively analyze the spatio-temporal characteristics of mixed-use vitality and carbon emissions across four distinct village typologies. The construction of mixed-use units, grounded in property rights and actual utilization, offers a unique lens for village examination and has proven effective in delineating the disparities among the four village categories:

(1) Different village typologies manifest marked differences in mixed-use intensity and carbon emissions, with the relationship between them also varying. E-commerce operation villages have the largest amount of units exhibiting high mixed-use intensity, whereas industrial production villages feature the most units with pronounced carbon emissions.

(2) Carbon emissions in rural contexts display clear typological structural traits: scattered distribution in traditional agricultural villages, core aggregation in industrial production villages, linear diminution in tourism service villages, and dissipative dispersal in E-commerce operation villages.

(3) Owing to the cyclical nature of industries, the volatility in carbon emissions varies significantly among the rural village categories, with tourism service villages registering the highest fluctuations and E-commerce operation villages the least.

While certain regions in China have outlined policies and regulations to foster the development of mixed-use villages, the practical implementation of mixed-use planning and construction remains in its infancy. In light of this, our study specifically proposes sustainable development pathways for mixed-use villages, focusing on eliminating "carbon islands", guiding "carbon flow", and inhibiting the growth of "carbon entropy".

Our research study has certain limitations, but it also makes a valuable contribution to the literature, paving the way for the further exploration of this topic in future studies, which could be carried out via the following approaches: (1) refining the computational evaluation model for mixed-use intensity and carbon emissions, (2) analyzing and quantifying the specific factors influencing carbon emissions in mixed-use villages, (3) investigating the applicability of the study's conclusions when extended to other regions. Moreover, due to space constraints, various details and data within this research study could not be thoroughly elaborated upon. Nonetheless, we hope this study can provide perspectives and insights for the low-carbon development of mixed-use rural areas in the Yangtze River Delta and broader regions in China.

**Author Contributions:** Conceptualization: Y.W. and H.Y.; methodology: Y.W. and C.Z.; investigation: Y.S. and X.W.; supervision: Y.L.; data curation: Y.W. and C.Z.; writing—original draft preparation: Y.W. and H.Y.; writing—review and editing: Y.W.; software: Y.S.; project administration: Y.L. All authors have read and agreed to the published version of the manuscript.

**Funding:** This research article was funded by the National Natural Science Foundation of China (No.51878593), the Center for Balance Architecture, Zhejiang University (No.KH-20212946), the Artificial Intelligence Key Technologies R&D Program of Hangzhou (2022AIZD0057), the Strategic Research and Consulting Project of Chinese Academy of Engineering (No.2022-XZ-51), the Scientific Research and Nurturing Foundation of Hangzhou City University (No.J-202308), and the Foundation of Hangzhou Federation of Humanities & Social sciences (No.2023HZSL-ZC011).

**Institutional Review Board Statement:** Not applicable.

**Informed Consent Statement:** Not applicable.

**Data Availability Statement:** Not applicable.

**Acknowledgments:** The authors would like to thank the editors and anonymous reviewers for their insightful comments and suggestions, which significantly contributed to enhancing the overall quality of this paper.

**Conflicts of Interest:** The authors declare no conflict of interest.

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
