# Peer review of "Spatial–Temporal Characteristics of Carbon Emissions in Mixed-Use Villages: A Sustainable Development Study of the Yangtze River Delta, China"

_sustainability, doi:10.3390/su152015060_

Round 1

Reviewer 1 Report

General Summary:

Thank you for giving me an opportunity to review the manuscript “Spatial-temporal characteristics of carbon emissions and sustainable development pathways in mixed-use villages: A case study of the Yangtze River Delta”. The authors investigated the relationship between mixed-use intensity and carbon emissions in rural areas of the Yangtze River Delta, intending to propose sustainable development pathways. After a careful reading of this article, I found that the manuscript requires moderate improvements to meet the publication criteria of the Sustainability MDPI journal. My specific comments are:

The title needs to be modified and shortened. It refers to spatio-temporal characteristics of sustainable development pathways too, which is not correct. Spatio-temporal characteristics are for carbon emissions only. Also, please add the country name in the title.

Abstract:

This section needs to be rewritten. No information about the study design and major numerical findings are reported here. The whole paragraph seems like an explanation of the problem taken by the authors and then each objective. No information about what this study yielded and its significance to currently available knowledge is justified.

Introduction

This section should be enriched with recent studies that addressed the problem individually or collectively. The linking of a problem with a hypothesis about the correlation between carbon emissions and SDG seems not very strong.

Also, It should provide a strong rationale for investigating carbon emissions and sustainable development pathways in mixed-use villages within the context of the Yangtze River Delta. Why this region was selected?

Results and discussion:

The manuscript does not adequately discuss the broader implications of the research beyond the study area. It should explore how the findings might be relevant to other regions facing similar challenges.

The discussion should be enriched with a sufficient number of supporting references. I can see authors made several claims in this section without proper justification and supporting literature. Not a single reference is available in this section.

Conclusion:

It looks like a repetition of the abstract and discussion. The conclusion should provide a clear summary of the key findings and their implications, and it should also suggest directions for future research in this area. It should leave the reader with a clear takeaway message from the study. Better to write it concisely within 250 words.

Reviewer 2 Report

The authors examined the relations between Yangtze River Delta's mixed-use villages and the rural area's carbon emissions.

Overall, the paper covers an important and necessary topic in analyzing micro-level CO2 emissions in rural China. However, I feel this paper needs to provide a more critical discussion related to the cross-disciplinary scope of the journal Sustainability.

The methods are adequately described. However, there is no proper discussion concerning the rural context in China and the updated literature. How significant are they? Most of the literature used is old and must be updated.

A few points must be better addressed:

Lines 48: I think it is necessary to discuss better the impact of China's dual-carbon strategy and its impacts on the mixed-use types of villages in rural China.

Line 84: I believe the theoretical background should be merged with the Introduction section.

Line 151: How was the sampling for choosing these villages made, considering the region has 3,100 villages in total? Why these eight are significant?

Line 167: Please better explain the various mixed-use rural types categorization.

Line 238: The temporal subset is not precise in the text.

Line 267: Figures 2 and 3 are the results.

Line 323: I suggest presenting the correlation values and not the classification of it. How significant are they?

Line 402: Again, the Discussion section regarding the updated literature available must be better addressed.

Reviewer 3 Report

General comments

I think it is an interesting article that offers great potential to advance in this line with innovative solutions applicable to real situations. At the same time, the application that made in this work is of great interest because it is necessary to have scientific studies that demonstrate the evidence, already contrasted, of the carbon emissions.

However, there are some questions that the authors could do to improve this manuscript. I make a few points in the specific suggestions that follow.

 Specific comments:

1)      I do not think it is necessary to put subparagraphs in the Introduction section. Although the objective of the authors has been to clarify the theoretical framework on which they base their research, it is possible to make a continuous narrative within the same section achieving good results with it.

2)      Can the authors provide some more statistical parameters in section 2.1 that allow the reader to contextualize more precisely the area of study? Logically, it is about providing information on the conditions of the study area in its potential for carbon emissions.

3)      Before making the calculation of the carbon footprint and the analysis proposed in section 3.3 it would be convenient that one of the first steps to follow was to specify clearly what are the limits established by the authors to evaluate emissions.

4)      In the following sections, the calculation of the carbon footprint according to scope for each of the activities mentioned above is addressed. In each case, the activity data required and the corresponding conversion and emission factors should be indicated.

5)      In order to understand the calculations, qualitative assessments for emissions have been established in the methodology described in section 3.4, specifically in Tables 1 and 2. Is it possible to establish the features of these valuations and their meaning? Are emission factors expressed in kg of CO2 equivalent considered? Therefore, in addition to CO2, they include the gases CH4 and N2O.

6)      I suggest a new wording of the Discussion section because it seems more a continuation of the explanation of the results than a comparison of the results obtained with those of similar studies, which should be cited, if necessary.

7)      The bibliographic references have not been completed with the DOI (Digital Object Identifier) indicator, which would be convenient for a more agile and truthful access of the cited publications. If it is not obligatory in this magazine, if it would be convenient to put it.

Minor editing of English language required.

Round 2

Reviewer 1 Report

All issues and concerns in the manuscript have been addressed in the revised version. I suggest accept in current form. Thank you.

Reviewer 2 Report

All my suggestions were adequately accommodated in this new version.

Reviewer 3 Report

Many thanks to the authors for addressing the suggestions made by this reviewer. My intention has been to clarify some aspects that help improve the better understanding of the manuscript.

Minor editing of English language required. Slight grammatical corrections are necessary.